# Measurement of Glutamate Suppression in a 6-OHDA-Induced Dopamine Deficiency Rat Model Following Acute Single-Dose L-DOPA Using GluCEST/MRS

**DOI:** 10.3390/biomedicines13112761

**Published:** 2025-11-12

**Authors:** Tensei Nakano, Kazuma Bono, Junpei Ueda, Masato Ohmi, Shigeyoshi Saito

**Affiliations:** 1Department of Medical Physics and Engineering, Division of Health Sciences, The University of Osaka Graduate School of Medicine, Suita 560-0871, Osaka, Japan; u354426k@ecs.osaka-u.ac.jp (T.N.); u996125b@ecs.osaka-u.ac.jp (K.B.); uedaj@sahs.med.osaka-u.ac.jp (J.U.); ohmi@sahs.med.osaka-u.ac.jp (M.O.); 2Department of Radiological Sciences, Faculty of Health Sciences, Morinomiya University of Medical Sciences, Osaka City 559-8611, Osaka, Japan; 3Department of Advanced Medical Technologies, National Cardiovascular and Cerebral Research Center, Suita 564-8565, Osaka, Japan; 4Immunology Frontier Research Center, The University of Osaka, Yamadaoka, Suita 565-0871, Osaka, Japan; 5World Premier International Research Center Initiative Premium Research Institute for Human Metaverse Medicine, The University of Osaka, Yamadaoka, Suita 565-0871, Osaka, Japan

**Keywords:** preclinical 7T-MRI/MRS, Parkinson’s disease, 6-OHDA-induced unilateral Parkinson’s disease rat model, glutamate, L-DOPA

## Abstract

**Background/Objectives**: The Glutamate Chemical Exchange Saturation Transfer (GluCEST) technique is an advanced imaging modality that enables non-invasive glutamate quantification using MRI. **Methods**: This study evaluated glutamate dynamics in Parkinson’s disease (PD) using a unilateral PD rat model, in which Wistar rats received 6-hydroxydopamine (6-OHDA) injections into the medial forebrain bundle, selectively eliminating dopaminergic neurons in the substantia nigra–striatum pathway. **Results**: The PD rat model exhibited a significant GluCEST increase (MTR Values: 3.0 ppm) compared to the sham-operated group, which was suppressed by administration of L-DOPA, a dopamine precursor drug (Sham: 0.9 ± 0.4%, PD: 2.0 ± 0.2%, Sham L-DOPA: 0.9 ± 0.5%, PD_L-DOPA: 0.8 ± 0.7%, *p* < 0.01). Additionally, magnetic resonance spectroscopy-derived glutamate data were consistent with GluCEST findings (Sham: 1.4 ± 0.03, PD: 1.7 ± 0.06, Sham_L-DOPA: 1.4 ± 0.12, PD_L-DOPA: 1.4 ± 0.10, *p* < 0.01). **Conclusions**: These findings suggest that GluCEST and magnetic resonance spectroscopy are valuable for assessing abnormal glutamate dynamics in the 6-OHDA-induced rat PD model. Furthermore, GluCEST may detect suppressed glutamate secretion following L-DOPA treatment, underscoring its potential for monitoring disease progression and therapeutic responses in PD.

## 1. Introduction

Parkinson’s disease (PD) is the second most common neurodegenerative disorder, affecting ≥6 million people worldwide. Its pathology is characterized by the progressive loss of neurons, primarily dopaminergic neurons connecting the substantia nigra to the striatum. The clinical diagnosis of PD relies on hallmark motor symptoms, including tremors, bradykinesia, muscle rigidity, and postural instability [1,2].

Beyond motor deficits resulting from dopaminergic neurodegeneration, neuropsychiatric symptoms, including dementia, frequently emerge during disease progression. A key contributor to these symptoms is glutamate dysregulation, specifically glutamate overexpression [3,4]. As the most abundant excitatory neurotransmitter in the mammalian brain, glutamate plays a crucial role in central neurotransmission [5]. Excessive glutamate levels overstimulate glutamate receptors, leading to intracellular Na^+^ and Ca^2+^ accumulation, oxidative stress, and direct neuronal damage, ultimately resulting in cell death [5,6]. Furthermore, electrical stimulation of DA neurons demonstrated co-transmission of glutamate (Glu) in DA neurons within the ventral striatum, supporting the relationship between dopamine and glutamate [7].To reproduce both the dopaminergic depletion underlying motor deficits and the glutamatergic imbalance relevant to non-motor features, we employed the 6-hydroxydopamine (6-OHDA) lesion model. 6-OHDA administered into the medial forebrain bundle (MFB) is a neurotoxin selectively targeting the dopaminergic system within the substantia nigra–striatum pathway. Once intracellularly absorbed, 6-OHDA undergoes oxidation, generating hydroxyl radicals that induce cellular damage [8,9]. This neurotoxin is widely used in preclinical PD research due to its ability to replicate striatal dopamine depletion and basal ganglia dysfunction, closely resembling the neurodegenerative processes observed in human PD [10,11,12]. L-DOPA is primarily administered to alleviate PD-related motor impairments [13], but it has also been shown to modulate glutamate metabolism [14]. Dopamine replenishment via L-DOPA significantly inhibits both hydroxyl radical formation and extracellular glutamate levels in the striatum, restoring the striatal DA-glutamate balance [8]. This effect suggests that L-DOPA not only improves motor function but also modulates non-motor circuits related to glutamatergic dysregulation, thereby providing an important rationale for its evaluation in the 6-OHDA model.

In 2012, Cai et al. introduced the chemical exchange saturation transfer (CEST) magnetic resonance imaging (MRI) technique, which led to the development of glutamate CEST (GluCEST), a non-invasive imaging method for quantifying brain glutamate levels using MRI [15]. Unlike magnetic resonance spectroscopy (MRS), CEST offers higher sensitivity and spatial resolution, enabling indirect glutamate detection and mapping in vivo through the assessment of metabolite-induced alterations in bulk water [16]. This technique quantifies glutamate indirectly by measuring the magnetization transfer ratio (MTR) at a specific frequency (3.0 ppm) [16]. GluCEST has been applied in clinical studies, including in investigations of epilepsy [17,18] and gliomas [19]. Elevated amide proton transfer CEST signals have also been reported in patients with PD [20,21]. In animal studies, GluCEST imaging has been performed in 1-methyl-4-phenyl-1,2,3,6-tetrahydropyridine (MPTP)-induced mouse PD models, where increased GluCEST signals in the striatum were observed [22]. However, GluCEST imaging has not yet been explored in a 6-OHDA-induced rat PD model. Currently, MRS is widely used in clinical settings to measure intratumoral glucose metabolism in cancer patients and brain metabolites in neurodegenerative diseases [23]. Identifying the origin of a metabolite’s chemical shift within the brain is straightforward, and highly accurate quantification is possible using dedicated software, such as LCmodel (Version 6.3-1L) [24]. MRS studies in MPTP animal models of PD have reported changes in striatal levels of Glu, and glutamine (Gln), γ-aminobutyric acid (GABA) along with their recovery following acute L-DOPA therapy [25]. In addition to Glu, MRS enables quantification of metabolites such as N-acetyl aspartate (NAA), creatine (Cr), and myo-inositol (mIns), and taurine (Tau). Thus, while GluCEST provides high-resolution mapping of glutamate, MRS offers highly specific quantification of multiple neuro metabolites, underscoring their complementarity in the study of PD.

This study aimed to assess GluCEST imaging for evaluating glutamate alterations in the brain in a 6-OHDA-induced unilateral rat PD model. Additionally, we investigated whether L-DOPA treatment modulated brain glutamate metabolism in this PD model.

## 2. Materials and Methods

### 2.1. Animals

All experimental protocols were approved by the Research Ethics Committee of Osaka University (R02-05-0). Animal procedures were conducted according to the Osaka University Guidelines for Animal Experimentation and the National Institutes of Health Guide for the Care and Use of Laboratory Animals. Male Wistar rats were obtained from Japan SLC (Hamamatsu, Japan), where model preparation was outsourced and conducted following SLC guidelines. All rats were housed in a controlled vivarium (24 °C, 12:12 h light/dark cycle) with regulated lighting, ad libitum feeding, and individual habitation after surgery until shipping. A total of 24 Wistar rats were employed. The breakdown was as follows; Sham: Pseudo-treatment model with saline administered to the medial forebrain bundle (n = 6), PD: 6-OHDA-induced unilateral Parkinson’s model rat (n = 6), Sham_L-DOPA: Rats treated with L-DOPA in a pseudo-treatment model in which saline was administered to the medial forebrain bundle (n = 6), PD_L-DOPA: 6-OHDA-induced unilateral Parkinson’s model rat treated with L-DOPA (n = 6).

### 2.2. 6-OHDA-Induced Unilateral PD Model

Reagents included 6-OHDA (MilliporeSigma, Darmstadt, Germany), desipramine, ascorbic acid, and apomorphine (FUJIFILM Wako Pure Chemical Corporation, Osaka, Japan). Model preparation required a stereotaxic retention device, electric clippers, drill, surgical scalpels, straight scissors, tweezers, wound openers, needle holder, micro syringe (10.5 μL), Aicom stereo guide, and microinjection cannula.

Animals were anesthetized with pentobarbital (50 mg/kg), dehaired from the occiput to the dorsal neck, and secured in a brain stereotaxic restraint system. A 3–4 cm skin incision was made, the connective tissue was excised, and the periosteum was carefully removed to expose the skull. Suture lines were verified, and the coordinates of the bregma and lambda were recorded. The administration site was localized at anteroposterior −4.4 mm, mediolateral −1.5 mm, and dorsoventral −7.8 mm relative to bregma and the dural surface. The skull was perforated with an electric drill, a microinjection cannula was inserted 7.8 mm below the brain surface, and 6-OHDA (2.25 mg/mL) with desipramine (25 mg/kg, i.p.) was administered (n = 6). The cannula remained in place for 1 min after administration. Afterward, antibiotics were given, and the incision was sutured. Rats injected with saline into the medial forebrain bundle (Mfb) served as sham controls (n = 6).

Behavioral evaluation with apomorphine was conducted. Two weeks after 6-OHDA administration (Figure 1), apomorphine (1 mg/kg, i.p.), a dopamine receptor agonist, was administered, and rotational movements were observed for approximately 30 min. As dopamine depletion in the right striatum induces leftward rotations, rats exhibiting >7 leftward rotations per minute were classified as successfully induced PD model animals, and MRI was performed.

### 2.3. L-DOPA Administration

L-DOPA (100 mg/kg) and benserazide (25 mg/kg), dissolved in saline, were administered intraperitoneally 10 min before MRI. The L-DOPA duration test followed the same procedure, with L-DOPA and benserazide administration. Experiments were conducted on 6 PD model (PD_L-DOPA: n = 6) and 6 sham rats (Sham_L-DOPA: n = 6).

L-DOPA aqueous solution was administered intraperitoneally at the same concentration and volume as used during MRI (L-DOPA,100 mg/kg; Benserazide Hydrochloride, 25 mg/kg), and rotational movements were assessed in a 37 cm × 48 cm observation box. Rotations were recorded over a 5 min period, with additional observations 2–3 min before and after. The average number of rotations per minute was calculated for each rat.

### 2.4. MRI Experiments

#### 2.4.1. MRI Equipment

MRI images were acquired from Wistar rats 6 weeks after 6-OHDA administration (Figure 1). Brain MRI was acquired using a horizontal 7-T scanner (PharmaScan 70/16 US; Bruker Biospin, Ettlingen, Germany) with a 40 mm inner diameter volume coil. Rats were positioned in a stereotaxic frame to minimize motion artifacts, with their mouths secured to prevent movement during imaging. Body temperature was maintained at 36.5 °C using regulated water flow and continuously monitored with a physiological monitoring system (SA Instruments Inc., Stony Brook, NY, USA). All MRI scans were performed under general anesthesia with isoflurane (3.0% for induction, 2.0% for maintenance) while maintaining a respiration rate of 50–80 breaths per minute.

#### 2.4.2. T_2_-Weighted Images

T_2_ weighted imaging (T_2_WI) scans acquired for coregistration with CEST imaging and MRS are shown. Axial T_2_WI was obtained using the Turbo Rapid Acquisition with Relaxation Enhancement (RARE) sequence with the following parameters: repetition time (TR)/echo time (TE) = 3200/33 ms, number of slices = 1, RARE factor = 8, number of averages = 2, field of view = 32 × 32 mm^2^, matrix size = 128 × 128, slice thickness = 1.0 mm, and scan time = 1 min 42 s.

#### 2.4.3. CEST

The CEST imaging pulse sequence consisted of a magnetization transfer (MT) RARE sequence modified to saturate various frequency offsets. CEST imaging of rat brains was performed using a RARE readout pulse sequence with a frequency-selective continuous wave saturation preparation pulse. The sequence parameters were as follows: field of view = 32 × 32 mm^2^, slice thickness = 1 mm, TR/TE = 3200/33, matrix size = 128 × 128, number of averages = 1, in-plane resolution = 250 × 250 μm^2^, 1 slice with thickness = 1 mm, and scan time = 29 min 0 s. The Z-spectra were acquired using a 3000-ms (50 ms × 60) saturation pulse at a B_1_ amplitude of 3.0 μT with frequency offsets from −4.8 to +4.8 ppm (step: 0.3 ppm, 32 images) and one S_0_ image (without MT pulse) [26,27,28,29]. A point-by-point B_0_ correction was applied from −1.0 to +1.0 ppm (step: 0.1 ppm, 21 images, scan time = 17 min 42 s) using a B_0_ map obtained via the water saturation shift referencing (WASSR) method [30]. Data acquired with 7-T MRI were analyzed and processed using MATLAB (R2020, MathWorks, Natick, MA, USA). A total of 54 images, including B_0_ mapping, were obtained.

The MTR values of the right striatum were analyzed using ImageJ 1.53t. The MTR asymmetry was calculated as described in a previous report [31]. The MTR asymmetry curves were generated from the CEST image series, and asymmetry maps were reconstructed at 0, 0.3, 0.6, 0.9, 1.2, 1.5, 1.8, 2.4, 2.7, 3.0, 3.3, 3.6, 3.9, and 4.2 ppm. The MT pulse parameters were as follows: continuous-wave saturation pulse shape = block pulse, length = 50 ms, number of pulses = 60, bandwidth = 25.6 Hz, and B_1_ amplitude = 3.0 µT.

#### 2.4.4. MRS

Magnetic field homogeneity was optimized using a fast, automated shimming technique via the mapping along projections (MAPSHIM) sequence, achieving good shimming within the volume of interest (between 8.9 and 12.1 Hz). MRS acquisition was performed using a point-resolved spectroscopy sequence (TR/TE = 2500/16 ms) with variable-power radio frequency pulses and optimized relaxation delay (VAPOR) water suppression. Metabolite spectra were obtained using 256 repetitions with VAPOR and 32 repetitions without VAPOR, yielding a total scan time of 10 min 40 s. Metabolite concentrations, including those of glutamine, glutamate, glutathione, myoinositol, N-acetyl aspartate, taurine, glycerophosphocholine + phosphocholine, creatine + phosphocreatine, and glutamate + glutamine, were quantified using the basic set of LCModel [32].

### 2.5. Histological Analysis

The primary antibodies used in this study were tyrosine hydroxylase (TH), specific to dopaminergic neurons, and glial fibrillary acidic protein (GFAP) to monitor the glial cell response. Six weeks postoperatively, after an MRI examination, rat brains were removed and fixed in 10% paraformaldehyde. The brain tissue was then embedded in paraffin wax, and 5 mm-thick sections were prepared. Sections were degreased with xylene and rehydrated using an ethanol/water wash. After rinsing, sections were incubated with hematoxylin for 5 min, followed by a 10 min rinse with warm water. The sections were then incubated with eosin for 3 min, rehydrated using an ethanol/water wash, and degreased with xylene. After 20 min, slides were mounted. TH and GFAP staining was performed under a fluorescence microscope (BZ-X810; KEYENCE CORPORATION, Osaka, Japan).

For image acquisition of TH- and GFAP-stained brain sections, brightness, color, and tonality were kept constant using the fluorescence microscope’s automatic mode. Images were converted to 8-bit in ImageJ 1.53a and binarized. Quantification was performed using ImageJ and averaged across the right cerebral hemisphere. The intensity of TH and GFAP immunostaining was quantified by placing multiple square ROIs in the right striatum and averaging the values. For quantification, images captured with a fluorescence microscope were converted to grayscale intensity. Thresholds for binarization were set at 100–120 for TH staining and 120–150 for GFAP staining before area percentage analysis.

### 2.6. Statistical Analysis

The MTR asymmetry and metabolite concentrations are expressed as mean ± standard deviation. Brain metabolite levels were standardized to Cr + PCr by dividing the quantitative values of each metabolite by the Cr + PCr value. Differences among the four groups, including the L-DOPA-treated group with MRI measurements, were analyzed using one-way analysis of variance with Tukey’s multiple comparison tests using GraphPad Prism software (Version 9; GraphPad Software, San Diego, CA, USA). Differences between PD and Sham rats in TH and GFAP staining were analyzed using an unpaired *t*-test. The threshold for statistical significance was set at *p* < 0.05. Notations for significant differences in the results graphs were defined as follows: * *p* < 0.05 and ** *p* < 0.01, *** *p* < 0.001.

## 3. Results

### 3.1. Model Induction Testing

Wistar rats injected with 6-OHDA into the right medial forebrain bundle (MFB) exhibited more than seven leftward rotations in response to apomorphine administration (Figure 2a). Therefore, the data obtained in this study (Figure 3, Figure 4, Figure 5, Figure 6 and Figure 7) were acquired from Wistar rats exhibiting PD-like symptoms.

When the effect of L-DOPA was tested, L-DOPA-induced rotational movements persisted for >120 min (Figure 2b). The duration of L-DOPA’s effect was monitored through rotational behavior, and the presented MRI data confirms that the imaging was performed during the period of L-DOPA efficacy.

### 3.2. CEST Imaging

Figure 3 presents MTR maps at 3.0 ppm (GluCEST), illustrating metabolic changes in glutamate. Figure 4 and Appendix A displays MTR values at each frequency from 0 to 4.2 ppm at 0.3 ppm intervals. The MTR at 3.0 ppm showed a significant increase in the PD model, which was suppressed by L-DOPA treatment (Sham: 0.9 ± 0.4%, PD: 2.0 ± 0.2%, Sham_L-DOPA: 0.9 ± 0.5%, PD_L-DOPA: 0. 8 ± 0.7%, *p* < 0.01).

### 3.3. MRS Study

Figure 5 presents the MRS spectrum acquired using MRS imaging with MRI. The MRS results (Figure 6 and Table 1) revealed an increase in glutamate in PD rats. This glutamate elevation was suppressed by L-DOPA (Sham: 1.4 ± 0.03, PD: 1.7 ± 0.06, Sham_L-DOPA: 1.4 ± 0.12, PD_L-DOPA: 1.4 ± 0.10, *p* < 0.01). A decrease in glutamine was observed in PD rats compared to Sham and Sham_L-DOPA rats (PD: 0.6 ± 0.05, Sham: 0.7 ± 0.04, Sham_L-DOPA: 0.7 ± 0.05, *p* < 0.01). Taurine levels were lower in PD rats compared to Sham_L-DOPA rats (PD: 1.0 ± 0.08, Sham_L-DOPA: 1.2 ± 0.06, *p* < 0.05).

### 3.4. Staining

TH staining evaluation confirmed that 6-OHDA selectively ablated dopaminergic neurons in the PD model (Sham: 56 ± 8.6%, PD: 2.7 ± 1.7%, *p* < 0.01) (Figure 7a–c). GFAP staining was conducted based on the hypothesis that the observed glutamate increase resulted from glial cell activation. However, GFAP staining revealed no significant increase in glial cell activation (Sham: 5.4 ± 1.9%, PD: 8.6 ± 2.8%) (Figure 7d–f).

## 4. Discussion

### 4.1. Quantitative Assessment of Glutamate Using CEST and MRS Techniques

The novelty of this study lies in the evaluation of glutamate metabolism using GluCEST and MRS in a 6-OHDA-induced unilateral rat PD model, particularly in combination with L-DOPA treatment. The PD rats exhibited a significant increase in GluCEST compared to Sham rats, and this increase was suppressed by L-DOPA (Figure 3). The GluCEST signal elevation in PD is consistent with preclinical studies using MPTP models [33] and aligns with glutamate quantification by the MRS study (Figure 6 and Table 1). Severe dopaminergic neuronal death induced by high doses of 6-OHDA and MPTP has been suggested to coexist with glutamate excitotoxicity. A 1 mM increase in glutamate in MRS raises MTR by 0.6%, aligning with this study’s results [15].

Compared with the more widely used amide proton transfer CEST [20,34], which is commonly applied in glioma imaging, a unique challenge for glutamate detection (GluCEST) is the fast exchange rate of the target amine proton at physiological temperature and pH [15]. Since the exchange rate of glutamate amine protons is approximately 2000 ± 500 s^−1^, low-field whole-body clinical scanners (B_0_ ≤ 3 T) do not meet this requirement [15]. Consequently, detecting glutamate via CEST requires B_0_ and B_1_ inhomogeneity correction, a high magnetic field (B0 ≥ 7 T), and high B_1_ power [16,35,36,37]. In human PD studies conducted at 3.0 T, increased MTR at 3.5 ppm has been reported, potentially reflecting abnormal protein accumulation such as α-synuclein [20,38]. While a B_1_ power of 2.0 μT is commonly used in APT CEST imaging, this study employed a higher B_1_ value of 3 μT, thereby minimizing the influence of amide protons [39]. In MRS, the only structural difference between glutamate and glutamine is in the carboxy and amide groups, necessitating static field strengths of 7 T or higher to differentiate these metabolites [40]. However, an important limitation of GluCEST in clinical applications is its specific absorption rate due to high magnetic fields (B0 ≥ 7 T) and B_1_ power [16].

CEST is valuable for metabolite mapping but has lower specificity per frequency than MRS. While a half-width of ~0.1 ppm is recommended for MRS, MT pulses in CEST imaging have a broader half-width, meaning the frequency band of interest includes multiple metabolites [24]. Additionally, factors such as pH and NOE (Nuclear Overhauser Enhancement) influence CEST effects [15,41]. Furthermore, since peaks arising from the chemical shift between bulk water and solute resonance may be influenced by conventional MT effects and direct saturation of bulk water near 0 ppm, MTR asymmetry analysis was employed. Accurate MTR asymmetry analysis requires correction for shifts in the bulk water resonance frequency, which was achieved using the WASSR method. In this study, WASSR was used to correct the Z-spectrum (Appendix A), and shimming was performed using B_0_ maps and MAPSHIM to minimize magnetic field inhomogeneity [30].

The VOI (Figure 5e) used for MRS data acquisition in this study was accurately positioned using T2WI. In addition, magnetic field inhomogeneity within the VOI (between 8.9 and 12.1 Hz) was confirmed to be minimal, ensuring high-precision data collection. The quantitative assessment using MRS suggests that, under this study’s imaging conditions, the MTR increase at 3.0 ppm primarily reflects a glutamate increase. The combination of MRS for highly specific quantification and CEST for whole brain mapping with high spatial resolution supports each technique’s utility, making it an essential tool for measuring brain metabolites.

### 4.2. Association Between Glial Cell Evaluation by GFAP Staining and Glutamate

The observed glutamate elevation in the PD region via GluCEST/MRS may be associated with glial cell activity [22,42,43]. Astrocytes, the primary glial cells, play a key role in glutamate reuptake and convert glutamate to glutamine via glutamine synthetase, thereby preventing excitotoxicity [42,44]. In PD, excessive glutamate levels, reduced activity of glutamine synthetase, and astrocytic hyperactivity have been reported. [45,46,47]. The GluCEST and MRS data reflect increased total glutamate levels, both intracellular and extracellular, and MRS also revealed decreased glutamine [48]. This imbalance may result from astrocytic dysfunction in PD. A decrease in taurine levels was observed in the PD model rats in this study (Figure 6 and Table 1). Similar to the imbalance between glutamate and glutamine, this change may be attributed to the loss of dopaminergic neurons affecting astrocytic function. Taurine is neuroprotective. MRS studies have consistently demonstrated reduced taurine levels in PD [49,50,51]. Dopamine D2 receptor signaling in astrocytes modulates extracellular taurine levels and excitatory neurotransmission in the dorsolateral striatum [52]. The findings suggest that the loss of dopaminergic neurons may trigger glutamate-induced neurotoxicity, which, in turn, may impact taurine levels. Taken together, taurine reduction may therefore reflect a loss of neuroprotective buffering against glutamate excitotoxicity. This positions taurine as an important mediator within the broader dopamine–glutamate imbalance in PD, where impaired taurine regulation exacerbates neuronal vulnerability and contributes to both motor and non-motor dysfunction.

However, the quantitative evaluation of GFAP staining did not indicate a significant increase in the PD model. The primary reason for this may be that glial cells are typically activated in response to physical damage to brain cells, and the surgical effects of 6-OHDA administration may have introduced variability [53]. This suggests limitations of the 6-OHDA model for PD compared to the MPTP model. GFAP staining specifically labels astroglia [54,55] and does not account for microglia. To achieve a more comprehensive assessment, Staining evaluations, including fluorescent staining [33] and IBA1 immunostaining [56,57] should be evaluated in a larger sample size. Studies of PD using animal models with 6-OHDA or MPTP typically focus on the subsequent observation of effects following dopamine neuron death. However, a limitation of this approach is the inability to account for the potential neurotoxicity of excessive glutamate, which may contribute to the initial dopamine neuron loss in PD [58].

### 4.3. Glutamate Dynamics Following L-DOPA Treatment in a Unilateral 6-OHDA Rat PD Model

Glutamate distribution in the brain varies depending on the clinical symptoms, the region analyzed, and the dosage of L-DOPA. In patients with PD who do not present with motor symptoms, neuropsychiatric disturbances such as dementia and depression have been reported to be associated with glutamatergic dysregulation [59,60]. GluCEST imaging in early PD may capture glutamate overexpression prior to full neurodegeneration, offering potential for early diagnosis and intervention. One month after unilateral lesioning of the nigrostriatal pathway in rats using 6-OHDA, extracellular glutamate in the striatum increases, followed by a decrease at three months, three months after 6-OHDA lesioning, extracellular striatal glutamate is decreased [61].

CEST enables the noninvasive mapping of metabolic changes throughout the brain. From the CEST images obtained in this study, we infer that metabolic changes occur not only in the right hemisphere, where 6-OHDA was administered but also throughout the brain. Thus, the affected hemisphere may influence the contralateral, ostensibly healthy hemisphere, potentially impacting whole-brain metabolism [62,63]. MRS studies in the prefrontal cortex have linked glutamate reduction to cognitive and behavioral impairments in both preclinical and clinical settings, highlighting regional differences in glutamate dynamics [64].

The GluCEST findings in this study reflect the effects of acute L-DOPA administration on the striatum following near-complete dopaminergic pathway loss induced by 6-OHDA. Dopamine and glutamate interact complementarily [65,66,67,68]. Low-dose L-DOPA administration significantly suppresses both extracellular glutamate levels and hydroxyl radical formation in the striatum, indicating a neuroprotective effect of L-DOPA [8]. This suggests that dopamine replenishment via L-DOPA temporarily modulates excessive glutamate. Conversely, repeated administration of high-dose L-DOPA has been shown to promote hydroxyl radical production in the striatum [69]. Chronic L-DOPA treatment may lead to glutamate overproduction associated with dyskinesia, which can be suppressed by glutamate receptor antagonists [70,71,72]. Therefore, it is suggested that the neuroprotective and toxic effects of L-DOPA may vary depending on the stage of damage to DA neurons, the amount of DA synthesized, and its rate of metabolism.

The advantage of 6-OHDA-treated PD animal models lies in the ease of model creation and the ability to confirm dopaminergic neuronal loss through TH staining and behavioral assessment with apomorphine [10,73] (Figure 2a and Figure 7a–c). The effects of apomorphine are limited, with negligible impact observed two weeks after administration [74]. Behavioral testing verifies that L-DOPA is acting on the brain during MRI testing (Figure 2b). The distribution of glutamate elevation in the brain is expected to vary depending on the extent of neuronal loss in the early stage, the timing of 6-OHDA administration and the dose of L-DOPA. 6-OHDA effectively and readily replicates various pathological features of PD. This model is anticipated to be a valuable tool for assessing glutamate dynamics using GluCEST in the future, and this study serves as a pioneering investigation in this area.

## 5. Conclusions

The MTR values at 3.0 ppm (GluCEST) were measured to indirectly quantify excessive glutamate secretion in a 6-OHDA-induced unilateral PD model rat. Furthermore, PD models treated with L-DOPA exhibited consistent suppression of glutamate levels, as detected by both GluCEST and MRS. This study is the first to apply GluCEST imaging to a 6-OHDA-induced PD model and uniquely combines it with L-DOPA treatment. GluCEST/MRS measurements in this model represent a promising approach for early detection of PD, evaluation of L-DOPA efficacy, and mitigation of dyskinesia.

## Figures and Tables

**Figure 1 biomedicines-13-02761-f001:**
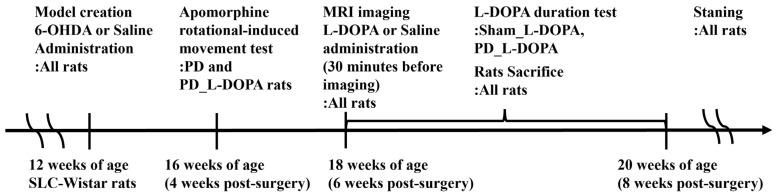
Schedule of the study. PD: 6-OHDA-induced unilateral Parkinson’s model rats (n = 6), Sham_L-DOPA: Rats treated with L-DOPA in a pseudo-treatment model in which saline was administered to the medial forebrain bundle (n = 6), PD_L-DOPA: 6-OHDA-induced unilateral Parkinson’s model rats treated with L-DOPA (n = 6).

**Figure 2 biomedicines-13-02761-f002:**
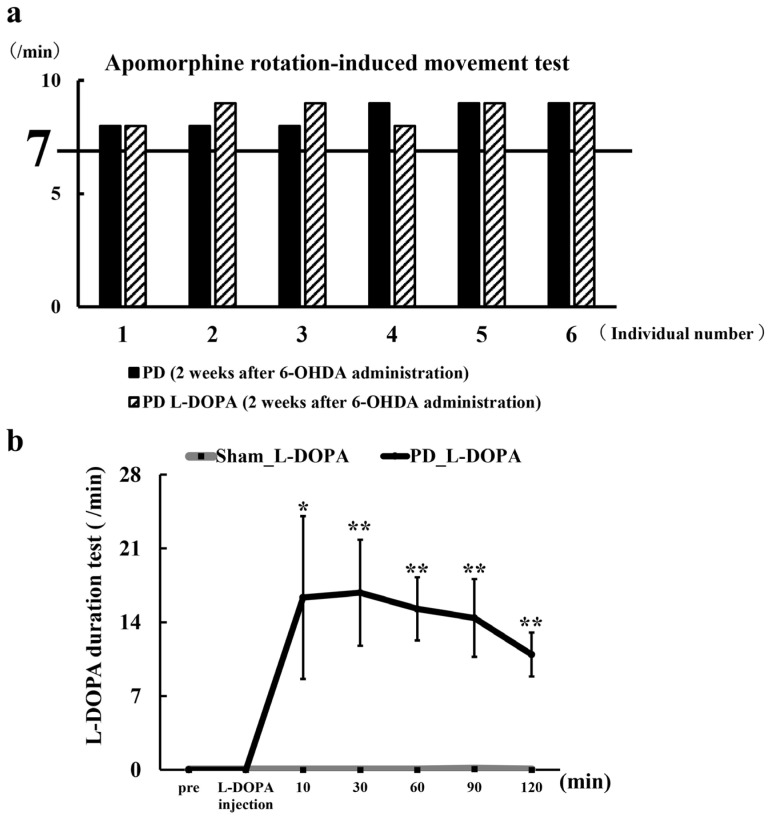
(**a**) Behavioral assessment using the dopamine receptor drug apomorphine. Two weeks after 6-OHDA administration, apomorphine (1 mg/kg, i.p.), a dopamine receptor agonist, was administered intraperitoneally, and rotational movements were observed for approximately 30 min after administration. (**b**) Graph of L-DOPA duration test. L-DOPA was administered intraperitoneally at the same concentration and dose used during MRI, and the number and timing of leftward rotations were recorded over time (L-DOPA;100 mg/kg, i.p, Benserazide Hydrochloride; 25 mg/kg, i.p). PD: 6-OHDA-induced unilateral Parkinson’s model rats (n = 6), Sham_L-DOPA: Rats treated with L-DOPA in a pseudo-treatment model in which saline was administered to the medial forebrain bundle (n = 6), PD_L-DOPA: 6-OHDA-induced unilateral Parkinson’s model rats treated with L-DOPA (n = 6). Comparisons between groups were analyzed using an unpaired *t*-test using (* *p* < 0.05, ** *p* < 0.01).

**Figure 3 biomedicines-13-02761-f003:**
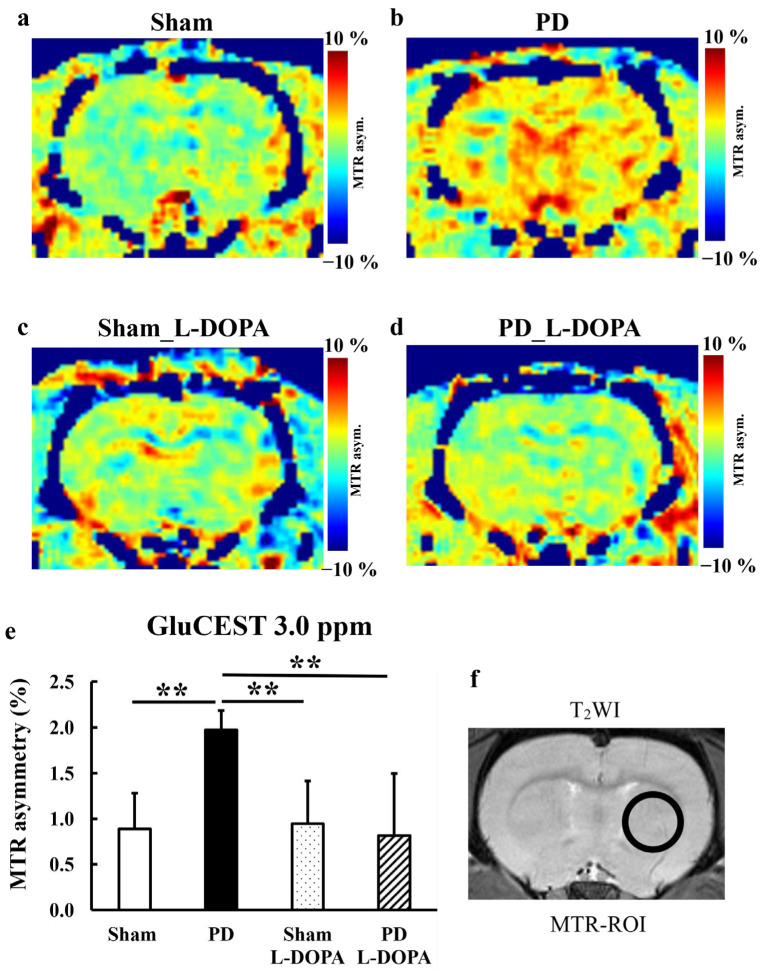
(**a**–**d**) Typical MTR maps for each group, (**e**) The MTR value at 3.0 ppm (GluCEST), (**f**) T_2_WI Image Showing Region of Interest for MTR. The PD rats exhibited a significant increase in GluCEST compared to Sham rats, and this increase was suppressed by L-DOPA. Comparisons between groups were analyzed using one-way analysis of variance with Tukey’s multiple comparison tests using (** *p* < 0.01). MTR: magnetization transfer ratio. Sham: Pseudo-treatment model with saline administered to the medial forebrain bundle (n = 6), PD: 6-OHDA-induced unilateral Parkinson’s model rats (n = 6), Sham_L-DOPA: Rats treated with L-DOPA in a pseudo-treatment model in which saline was administered to the medial forebrain bundle (n = 6), PD_L-DOPA: 6-OHDA-induced unilateral Parkinson’s model rats treated with L-DOPA (n = 6).

**Figure 4 biomedicines-13-02761-f004:**
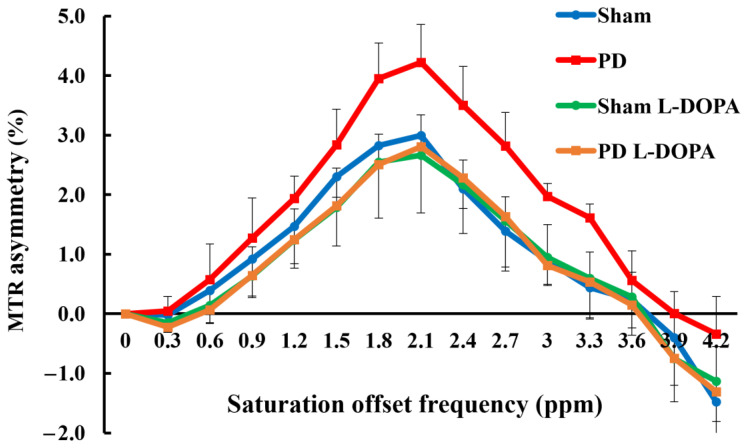
Cumulative MTR asymmetry curves. Comparisons between groups were analyzed using one-way analysis of variance with Tukey’s multiple comparison tests. MTR: magnetization transfer ratio. Sham: Pseudo-treatment model with saline administered to the medial forebrain bundle (n = 6), PD: 6-OHDA-induced unilateral Parkinson’s model rats (n = 6), Sham_L-DOPA: Rats treated with L-DOPA in a pseudo-treatment model in which saline was administered to the medial forebrain bundle (n = 6), PD_L-DOPA: 6-OHDA-induced unilateral Parkinson’s model rats treated with L-DOPA (n = 6).

**Figure 5 biomedicines-13-02761-f005:**
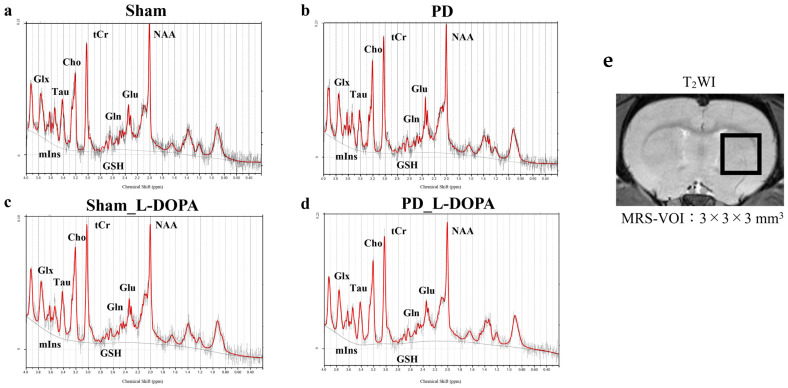
MRS of all groups. (**a**–**d**) MR spectra measured. (**e**) T_2_WI image showing voxel of interest for MRS. Using T_2_WI images in three orthogonal planes, a 3 × 3 × 3-mm^3^ volume of interest was positioned within the right striatum. Sham: Pseudo-treatment model with saline administered to the medial forebrain bundle (n = 6), PD: 6-OHDA-induced unilateral Parkinson’s model rat (n = 6), Sham_L-DOPA: Rats treated with L-DOPA in a pseudo-treatment model in which saline was administered to the medial forebrain bundle (n = 6), PD_L-DOPA: 6-OHDA-induced unilateral Parkinson’s model rat treated with L-DOPA (n = 6). MRS: Magnetic resonance spectroscopy.

**Figure 6 biomedicines-13-02761-f006:**
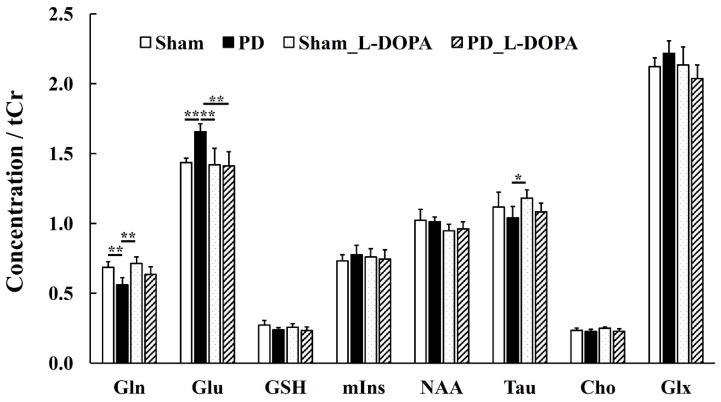
MRS metabolites/tCr. The Graph of MRS metabolites/tCr in the right Mfb 6 weeks after 6-OHDA administration. Glutamine (Gln), glutamate (Glu), glutathione (GSH), myoinositol (mIns), N-acetyl aspartate (NAA), taurine (Tau), glycerophosphocholine + phosphocholine (Cho), creatine + phosphocreatine (tCr), and glutamine + glutamate (Glx). Comparisons between groups were analyzed using one-way ANOVA with Tukey’s multiple comparison tests using Prism 9 (* *p* < 0.05, ** *p* < 0.01). Sham: Pseudo-treatment model with saline administered to the medial forebrain bundle (n = 6), PD: 6-OHDA-induced unilateral Parkinson’s model rat (n = 6), Sham_L-DOPA: Rats treated with L-DOPA in a pseudo-treatment model in which saline was administered to the medial forebrain bundle (n = 6), PD_L-DOPA: 6-OHDA-induced unilateral Parkinson’s model rat treated with L-DOPA (n = 6). MRS: Magnetic resonance spectroscopy.

**Figure 7 biomedicines-13-02761-f007:**
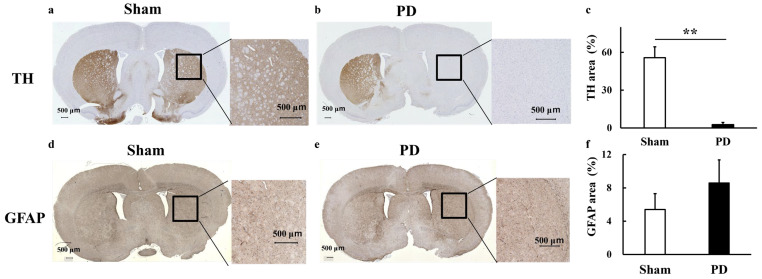
Histological experiments. (**a**–**c**) Images and graphs of TH staining and (**d**–**f**) GFAP staining. TH staining confirmed that the PD model in this study exhibited the destruction of dopaminergic neurons. GFAP staining revealed no significant increase in glial cell activation. Comparisons between groups were analyzed using an unpaired *t*-test using Prism 9 (** *p* < 0.01). Sham: Pseudo-treatment model with saline administered to the medial forebrain bundle (n = 6), PD: 6-OHDA-induced unilateral Parkinson’s model rats (n = 6), Sham_L-DOPA: Rats treated with L-DOPA in a pseudo-treatment model in which saline was administered to the medial forebrain bundle (n = 6), PD_L-DOPA: 6-OHDA-induced unilateral Parkinson’s model rats treated with L-DOPA (n = 6). TH: Tyrosine hydroxylase, GFAP: Glial fibrillary acidic protein.

**Table 1 biomedicines-13-02761-t001:** Metabolite concentration ratios measured by MRS.

/tCr	Sham	PD	Sham_L-DOPA	PD_L-DOPA
Gln	0.7 ± 0.04 **	0.6 ± 0.05	0.7 ± 0.05 **	0.6 ± 0.06
Glu	1.4 ± 0.03 **	1.7 ± 0.06	1.4 ± 0.12 **	1.4 ± 0.10 **
GSH	0.3 ± 0.03	0.2 ± 0.01	0.3 ± 0.03	0.2 ± 0.03
mIns	0.7 ± 0.05	0.8 ± 0.07	0.8 ± 0.06	0.7 ± 0.07
NAA	1.0 ± 0.08	1.0 ± 0.03	0.9 ± 0.05	1.0 ± 0.05
Tau	1.1 ± 0.11	1.0 ± 0.08	1.2 ± 0.06 *	1.1 ± 0.06
Cho	0.2 ± 0.02	0.2 ± 0.02	0.2 ± 0.01	0.2 ± 0.02
Glx	2.1 ± 0.06	2.2 ± 0.09	2.1 ± 0.13	2.0 ± 0.10

Abbreviations: glutamine (Gln), glutamate (Glu), glutathione (GSH), myoinositol (mIns), *N*-acetyl aspartate (NAA), taurine (Tau), glycerophosphocholine + phosphocholine (Cho), creatine + phosphocreatine (tCr), glutamine + glutamate (Glx). Sham: Pseudo-treatment model with saline administered to the medial forebrain bundle (n = 6), PD: 6-OHDA-induced unilateral Parkinson’s model rats (n = 6), Sham_L-DOPA: Rats treated with L-DOPA in a pseudo-treatment model in which saline was administered to the medial forebrain bundle (n = 6), PD_L-DOPA: 6-OHDA-induced unilateral Parkinson’s model rats treated with L-DOPA (n = 6). MRS: Magnetic resonance spectroscopy. Comparisons between groups were analyzed using one-way analysis of variance with Tukey’s multiple comparison tests (vs. PD, * *p* < 0.05, ** *p* < 0.01).

## Data Availability

The datasets generated during and analyzed during the current study are available from the corresponding author on request.

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
