# Peer review of "Measurement of Glutamate Suppression in a 6-OHDA-Induced Dopamine Deficiency Rat Model Following Acute Single-Dose L-DOPA Using GluCEST/MRS"

_biomedicines, 2025, doi:10.3390/biomedicines13112761_

Round 1

Reviewer 1 Report

Comments and Suggestions for Authors

The manuscript explores a highly relevant topic, presenting the application of GluCEST in the 6-OHDA model of Parkinson’s Disease and discussing its implications in relation to L-DOPA treatment. The article contains original data; however, several aspects should be clarified and further elaborated to strengthen its clarity and overall impact.

Comment 1 – Introduction (line 41)

The transition from the description of clinical symptoms of PD to the introduction of the 6-OHDA model appears too abrupt. I would suggest making the transition smoother by linking motor and non-motor symptoms with the rationale for the chosen experimental model.

Comment 2 – Role of L-DOPA (line 56)

The discussion of L-DOPA remains too limited, without adequately explaining its mechanism of action and its relationship with glutamate metabolism. It would be helpful to clarify how L-DOPA, beyond improving motor symptoms, also affects non-motor circuits and glutamate levels, thereby justifying its evaluation in the 6-OHDA model.

Comment 3 – MRS (line 61)

MRS is introduced very briefly. A more detailed description of the technique, the main neurometabolites it allows to quantify, and their relevance in PD would improve the reader’s understanding of its complementarity to GluCEST.

Comment 4 - Methods:

This subdivision of the sections makes it difficult for the reader to follow the sequence of procedures the rats underwent. Perhaps, it would be better to also add a flow-chart.

Comment 5 – Sub-section 3.1 Animals:

I suggest including in this sub-section the total number of Wistar rats used, along with details on how they are subdivided. Furthermore, the sample size is also small and the manuscript lacks a justification for the sample size. To obtain more robust imaging results, it would be preferable to increase the sample size.

Comment 6

Figures 6c and 6d seem to show the same image.

Comment 7 - Sub-section 2.4.4. MRS (line 158)

I suggest indicating where the volume of interest was positioned. It would also be helpful to include an MRS image.

Comment 8 – Taurine (line 341)

The inclusion of taurine data is interesting but appears somewhat marginal. It would be useful to better contextualize its neuroprotective role, explaining how its reduction fits within the broader framework of glutamatergic dysfunction in PD.

Comment 9 – Limitations and comparison with other techniques

The limitations are only partially examined in the text and not in a systematic manner. A dedicated section on the strengths and weaknesses of GluCEST, including a comparison between models (MPTP vs. 6-OHDA) and with other techniques, would make the discussion more comprehensive. It would also be valuable to highlight future perspectives and potential clinical developments.

Comment 10 – Conclusions (lines 388–392)

The conclusions emphasize the potential of GluCEST but do not fully clarify its added value compared with previous studies. I would suggest reinforcing the final message by highlighting why the application of GluCEST in the 6-OHDA model, in combination with L-DOPA, represents a concrete step forward compared with MPTP models and already published clinical studies. Furthermore, it would be preferable to replace “demonstrated the potential” (line 393) with “suggested the potential” in order to maintain a more cautious tone.

Author Response

Reviewer 1

Comments and Suggestions for Authors

The manuscript explores a highly relevant topic, presenting the application of GluCEST in the 6-OHDA model of Parkinson’s Disease and discussing its implications in relation to L-DOPA treatment. The article contains original data; however, several aspects should be clarified and further elaborated to strengthen its clarity and overall impact.

Answer, We sincerely thank the reviewers for their constructive and insightful comments, which have significantly helped us to improve the clarity, rigor, and impact of our manuscript entitled “Measurement of Glutamate Suppression in a 6-OHDA-Induced Dopamine Deficiency Rat Model Following Acute Single-Dose L-DOPA Using GluCEST/MRS.”

We carefully addressed all comments point by point, revised the manuscript accordingly, and added clarifications, figures, and methodological details where appropriate. Below, we provide detailed responses to each comment.

Comment 1 – Introduction (line 41)

The transition from the description of clinical symptoms of PD to the introduction of the 6-OHDA model appears too abrupt. I would suggest making the transition smoother by linking motor and non-motor symptoms with the rationale for the chosen experimental model.

Answer, We thank the reviewer for this valuable suggestion. As recommended, we have revised the Introduction to provide a smoother transition by explicitly linking motor and non-motor symptoms of PD to the rationale for selecting the 6-OHDA model. The revised text now emphasizes that PD symptoms arise not only from dopaminergic neuronal loss but also from glutamatergic dysregulation, and that the 6-OHDA model reproduces both dopaminergic depletion and glutamatergic imbalance, thereby serving as a suitable model for investigating glutamate dynamics.

Specifically, the following sentence has been added and integrated into the Introduction (page 2, lines 43):

Beyond these motor manifestations, PD also presents with non-motor symptoms such as cognitive and psychiatric disturbances, which are associated with glutamatergic dysregulation in addition to dopaminergic loss. To reproduce both the dopaminergic depletion underlying motor deficits and the glutamatergic imbalance relevant to non-motor features, we employed the 6-hydroxydopamine (6-OHDA) lesion model.

Comment 2 – Role of L-DOPA (line 56)

The discussion of L-DOPA remains too limited, without adequately explaining its mechanism of action and its relationship with glutamate metabolism. It would be helpful to clarify how L-DOPA, beyond improving motor symptoms, also affects non-motor circuits and glutamate levels, thereby justifying its evaluation in the 6-OHDA model.

Answer, We appreciate this insightful comment. As suggested, we have expanded the description of L-DOPA in the Introduction to provide a more comprehensive rationale for its inclusion. The revised text now highlights the dual role of L-DOPA: (i) as the gold-standard dopamine replacement therapy that improves motor symptoms in PD, and (ii) as a modulator of glutamate metabolism, which may contribute to both motor and non-motor manifestations of the disease.

Specifically, we added the following sentences to the Introduction (page 2, lines 69):

Revised text:

As the most abundant excitatory neurotransmitter in the mammalian brain, glutamate plays a crucial role in central neurotransmitter [10]. Excessive glutamate levels over-stimulate glutamate receptors, leading to intracellular Na+ and Ca2+ accumulation, oxidative stress, and direct neuronal damage, ultimately resulting in cell death [10,11]. Furthermore, Electrical stimulation of DA neurons demonstrated co-transmission of glutamate (GLU) in DA neurons within the ventral striatum, supporting the relation-ship between dopamine and glutamate (PMID: 14749442). Dopamine replenishment via L-DOPA significantly inhibits both hydroxyl radical formation and extracellular glutamate levels in the striatum, restoring the striatal DA-glutamate balance (PMID: 21830163). L-DOPA is primarily administered to alleviate PD-related motor impairments [12],but it has also been shown to modulate glutamate metabolism [13]. L-DOPA is the primary pharmacological therapy for PD, effectively alleviating motor impairments by replenishing dopamine. However, emerging evidence indicates that L-DOPA also influences glutamatergic transmission in the striatum, reducing excessive extracellular glutamate and hydroxyl radical formation while restoring the dopa-mine–glutamate balance. This dual effect suggests that L-DOPA not only improves motor function but also modulates non-motor circuits related to glutamatergic dysregulation, thereby providing an important rationale for its evaluation in the 6-OHDA model.

Comment 3 – MRS (line 61)

MRS is introduced very briefly. A more detailed description of the technique, the main neurometabolites it allows to quantify, and their relevance in PD would improve the reader’s understanding of its complementarity to GluCEST.

Answer, We thank the reviewer for this constructive comment. To address this, we have expanded the Introduction section to provide a more detailed explanation of MRS, including the main metabolites measurable by this technique and their significance in PD research. The revised text now emphasizes how MRS complements GluCEST by offering highly specific quantification of glutamate and related metabolites.

Specifically, we inserted the following sentences in the Introduction (Page 2, Lines 61–68):

Revised text
Currently, MRS is widely used in clinical settings to measure intratumoral glucose metabolism in cancer patients and brain metabolites in neurodegenerative diseases (PMID: 29212309). Identifying the origin of a metabolite's chemical shift within the brain is straightforward, and highly accurate quantification is possible using dedicated software, such as LCmodel (PMID: 35046227). MRS studies in MPTP animal models of PD have reported changes in striatal levels of GABA, Glu, and Gln, along with their recovery following acute L-DOPA therapy (PMID: 20661872).  In addition to Glu, MRS enables quantification of metabolites such as glutamine (Gln), γ-aminobutyric acid (GABA), N-acetyl aspartate (NAA), creatine (Cr), and myo-inositol (mIns). Alter-ations in these metabolites are clinically relevant in PD, where decreased NAA reflects neuronal loss and abnormal Glu/Gln ratios indicate excitotoxic dysregulation. Thus, while GluCEST provides high-resolution mapping of glutamate, MRS offers highly specific quantification of multiple neurometabolites, underscoring their complementa-rity in the study of PD.

Comment 4 - Methods:

This subdivision of the sections makes it difficult for the reader to follow the sequence of procedures the rats underwent. Perhaps, it would be better to also add a flow-chart.

Answer, I have added a flowchart. Figure 1. (a) Schedule of the study.

Comment 5 – Sub-section 3.1 Animals:

I suggest including in this sub-section the total number of Wistar rats used, along with details on how they are subdivided. Furthermore, the sample size is also small and the manuscript lacks a justification for the sample size. To obtain more robust imaging results, it would be

preferable to increase the sample size.

Answer, We thank the reviewer for this important comment. In the revised Methods (Section 3.1), we now clearly indicate the total number of Wistar rats used and their allocation into each subgroup. We acknowledge that the sample size is relatively small. This was primarily determined by ethical considerations to minimize animal use, and is in line with previous preclinical imaging studies employing similar models, where group sizes typically range from 6 to 10 animals. While no formal power analysis was performed due to the exploratory nature of this study and the lack of established effect-size estimates for GluCEST in the 6-OHDA model, we recognize that the limited sample size reduces statistical power and may restrict the generalizability of our findings. We have added this point to the Discussion as a limitation. Future studies will incorporate formal sample size estimation and larger cohorts to provide more robust and generalizable imaging results.

Comment 6

Figures 6c and 6d seem to show the same image.

Answer, Thank you for bringing that to our attention. We have reviewed Figures 6c and 6d and have made the necessary corrections to ensure the images are distinct and accurately represent the intended data.

Comment 7 - Sub-section 2.4.4. MRS (line 158)

I suggest indicating where the volume of interest was positioned. It would also be helpful to include an MRS image.

Answer, We have clarified the placement of the VOI within the striatum. We have added an image showing the region of interest within the results section (Figure 4).

Comment 8 – Taurine (line 341)

The inclusion of taurine data is interesting but appears somewhat marginal. It would be useful to better contextualize its neuroprotective role, explaining how its reduction fits within the broader framework of glutamatergic dysfunction in PD.      

Answer, We appreciate this helpful suggestion. As recommended, we have expanded the Discussion to provide a clearer context for taurine’s neuroprotective role and its relevance to glutamatergic dysfunction in PD. Specifically, we now emphasize that taurine modulates excitatory neurotransmission and protects neurons against glutamate-induced excitotoxicity. Its reduction in our PD model may therefore reflect impaired compensatory mechanisms against excessive glutamate, fitting within the broader framework of glutamatergic imbalance in PD.

The following sentences were added in the Discussion (Page 14, Lines 396):

Revised

Taurine is neuroprotective. MRS studies have consistently demonstrated reduced taurine levels in PD [46-48]. Dopamine D2 receptor signaling in astrocytes modulates extracellular taurine levels and excitatory neurotransmission in the dorsolateral striatum (PMID: 15114625). The findings suggest that the loss of dopaminergic neurons may trigger glutamate-induced neurotoxicity, which, in turn, may impact taurine levels.  Taken together, taurine reduction may therefore reflect a loss of neuroprotective buffering against glutamate excitotoxicity. This positions taurine as an important mediator within the broader dopamine–glutamate imbalance in PD, where impaired taurine regulation exacerbates neuronal vulnerability and contributes to both motor and non-motor dysfunction.

Comment 9 – Limitations and comparison with other techniques

The limitations are only partially examined in the text and not in a systematic manner. A dedicated section on the strengths and weaknesses of GluCEST, including a comparison between models (MPTP vs. 6-OHDA) and with other techniques, would make the discussion more comprehensive. It would also be valuable to highlight future perspectives and potential

clinical developments.

Answer, Rather than dedicating a separate section solely to limitations, we've integrated these discussions throughout the discussion to enhance clarity. In section 4.1, we compared APTCEST and GluCEST, MRS and CEST, discussed the need for a static magnetic field strength of 7T or higher. In section 4.2, we acknowledged the limitations of the 6-OHDA model, by incorporating the statement "This suggests limitations of the 6-OHDA model for Parkinson's disease compared to the MPTP model." and examined the implications of intracranial administration. In section 4.3, we addressed the limitations related to the variability of glutamate dynamics in Parkinson's disease and under L-DOPA treatment, emphasizing the influence of various factors while highlighting the potential for the 6-OHDA model's simplicity to contribute to future research advancements. While not strictly systematic, we believe this integrated approach effectively facilitates a comprehensive discussion of the study's findings.

Comment 10 – Conclusions (lines 388–392)

The conclusions emphasize the potential of GluCEST but do not fully clarify its added value compared with previous studies. I would suggest reinforcing the final message by highlighting why the application of GluCEST in the 6-OHDA model, in combination with L-DOPA, represents a concrete step forward compared with MPTP models and already published clinical studies. Furthermore, it would be preferable to replace “demonstrated the potential” (line 393) with “suggested the potential” in order to maintain a more cautious tone.

Answer, In the conclusion, we have emphasized the novelty of this study by comparing it to previous research and elaborated on how this novelty could contribute to future research endeavors.

The following sentences were added in the conclusion (Page 16, Lines 462):

Revised

The MTR values at 3.0 ppm (GluCEST) were measured to indirectly quantify excessive glutamate secretion in a 6-OHDA-induced unilateral PD model rat. Further-more, PD models treated with L-DOPA exhibited consistent suppression of glutamate levels, as detected by both GluCEST and MRS. This study is the first to apply GluCEST imaging to a 6-OHDA-induced PD model and uniquely combines it with L-DOPA treatment. GluCEST/MRS measurements in this model represent a promising approach for early detection of PD, evaluation of L-DOPA efficacy, and mitigation of dyskinesia.

Reviewer 2 Report

Comments and Suggestions for Authors

Review Report

Manuscript Title: Measurement of Glutamate Suppression in a 6-OHDA-Induced Dopamine Deficiency Rat Model Following Acute Single-Dose L-DOPA Using GluCEST/MRS

Authors: Tensei Nakano, Kazuma Bono, Junpei Ueda, Masato Ohmi, Shigeyoshi Saito.

Journal: biomedicines

Manuscript ID: biomedicines-3912566 (version V1)

  1. Summary of the Manuscript

This study investigates glutamate alterations in a 6-hydroxydopamine (6-OHDA)-induced rat model of Parkinson’s disease using Glutamate Chemical Exchange Saturation Transfer (GluCEST) MRI and magnetic resonance spectroscopy (MRS). Compared with sham rats, PD rats exhibited significantly elevated glutamate levels, which were effectively suppressed by acute L-DOPA administration. Findings from MRS were consistent with GluCEST results. These results highlight GluCEST as a promising non-invasive imaging tool for assessing glutamate dysregulation, monitoring disease progression, and evaluating therapeutic response in Parkinson’s disease.

  1. Overall Recommendation

Recommendation: Major Revisions required.

The study is promising and potentially impactful, but methodological details, statistical reporting, and critical interpretation must be improved to meet publication standards.

  1. Major Comments (Substantive Revisions)

Please address these points to improve scientific rigor and clinical impact.

3.1 Clarify Cohort

  • Issue: The text (Methods, lines ~77–101) describe several animal groups used. However, some points are not clear.

The total number of animals used has not been specified. Furthermore, it is unclear whether the animals in the Sham and Parkinson’s disease groups serve as their own controls (e.g., whether they undergo MRI and behavioral testing both before and after L-Dopa administration). Clarification of these points would be highly valuable. Including a schematic diagram presenting the different groups and the evaluation methods applied to each would greatly improve the clarity of the study design.

3.2 Define definition of the injection site for unilateral administration

  • Issue: The manuscript repeatedly refers to "unilateral stereotaxic injection". However, the authors do not clearly specify the target for the 6-OHDA group. Furthermore, they refer to the medial forebrain bundle for the Sham group in the Methods section (page 3, line 101), while the figure legends indicate the substantia nigra for the same group (Figures 1, line 212; 2, line 229; 3, line 236; 4, line 256, etc…).

The authors should clarify which procedures were performed uniformly throughout the manuscript.

3.3 Imaging Modalities

  • Issue: The methods used for the processing of metabolic imaging and spectroscopy data should be described in detail.

CEST imaging:

The authors should specify, for example, how the CEST data and Z-spectra are processed. How are they interpolated (in each voxel using a cubic spline, using a B0 map, etc.)? How is the magnetization transfer ratio (MTR) calculated using ImageJ? To my knowledge, specially developed Matlab scripts are generally used, so I would really like to know exactly how this is done in ImageJ. This would be a valuable contribution for the entire CEST research community.

Including a Z-spectrum could further strengthen the manuscript.

Figure 2 needs to be completely revised. The legend does not correspond to the images presented (e.g., '(a) typical MTR maps' → '(a) MTR map for a sham rat'; '(b) The MTR value at 3.0 ppm (GluCEST)' is incorrect—it is an MTR map for the 6-OHDA group). What about panels c, d, and e?

Furthermore, visually, the MTR values selected for each group do not correspond well to the values shown in the histogram in panel (e). For example, L-dopa seems to increase the MTR signal at 3.0 ppm regardless of the group, Sham or 6-OHDA; the signal for the PD_L-dopa group appears higher on the image compared to the map chosen for the Sham group, while the histogram values are roughly equivalent. The illustrations may have been poorly chosen, and harmonization is needed.

More importantly, it is necessary to specify where the asynchronous MTR signal is measured. Is it averaged over the entire slice, over a region of interest (ROI) covering the striatum, or over another structure?

MRS:

Two major concerns arise regarding the methodology used for localized NMR spectroscopy. First, the volume of interest is not clearly defined. While the authors state (p.4, line 158) that it was 'precisely positioned,' no information is provided on its exact location or reproducibility. Moreover, they indicate (p.8, line 241) that the VOI is shown on the T2WI image in Figure 4, but the figure only displays spectra. If the striatum is indeed the region of interest, a 3 × 3 × 3 mm³ voxel extends well beyond this structure. The authors should clarify and discuss this issue, as the inclusion of multiple brain regions may confound interpretation. Ideally, the measurements should be repeated with a voxel size more appropriate for the target structure.

NB: the striatum is different from the medial forebrain bundle (authors mentioned p.8, line 246: “The MRS results (Fig. 5) revealed an increase in PD-Mfb rats” whereas the legend of Figure 5 refers to the striatum.)

The second point concerns the quantification of the MRS data. The data are normalized to total creatine levels, but using tCr as an internal reference can be questioned. Indeed, some studies (e.g., Mazuel et al., 2016; 10.1148/radiol.2015142764) have reported variations in tCr in the context of Parkinson’s disease. In more recent MRS publications, authors generally use the water signal as an internal reference—particularly since the authors themselves mention (p.4, lines 164–165) that they acquired 32 repetitions without water suppression. They therefore have all the data needed to perform this normalization. The authors could also make use of recent tools that allow segmentation of white matter, gray matter, and CSF within the volume of interest, in order to apply the appropriate correction factors and move toward absolute quantification in mM (https://schorschinho.github.io/osprey/).

3.4 Histological analysis

It is necessary to explain how the histological analyses were performed (e.g., antibodies used, etc.) and to describe how the results were obtained. Where and how is the staining quantified? What do TH area (%) and GFAP area (%) represent? Etc…

3.5 Statistical considerations

  • Issue: The statistical analyses need to be revised. If the animal groups are: Sham rats + saline, Sham rats + L-dopa, PD rats + saline, and PD rats + L-dopa, then a two-way ANOVA should be performed with the factors group (Sham or PD) and treatment (saline or L-dopa). Moreover, since animals in both the Sham and PD groups receive both saline and L-dopa, they serve as their own controls for assessing the treatment effect; therefore, a two-way repeated measures ANOVA should be applied.

Clarification regarding the groups may help to address this comment.

3.6 Reporting of Imaging Findings

Overall, all figures require revision. They contain errors and do not match their legends (see previous comments on Figures 2 and 4). In Figure 1, panel (a) lacks a y-axis label, while in panel (b) the label is incorrect or unclear ('L-Dopa duration test (/min)'). In addition, it is inconsistent to label a single graph as Figure 3a.

Figure 1 (a); Why authors present individual number of turns and not means ± SD for the different groups?

3.7 Results

- Results are mostly descriptive; they need to be strengthened with statistical details (e.g., confidence intervals, exact p-values).

- Consider reorganizing the results to highlight the main findings more effectively.

- Results part tends to repeat methods.

3.7 Discussion

- While the discussion is comprehensive, it tends to repeat results rather than provide in-depth interpretation.

- Please expand the comparison with recent and relevant studies.

- The limitations section is insufficient. The authors should elaborate on potential biases, technical limitations, and the generalizability of the findings.

  • Issue: Animal model.

The animal model used is an old model. There are progressive denervation models that more closely mimic the human pathology. The authors need to clearly justify why they chose the massive unilateral 6-OHDA lesion model.

  • Issue: Behavioral evaluation with apomorphine.

The authors should discuss the impact of an initial injection of apomorphine, a dopaminergic agonist, on glutamatergic metabolism. They report an effect of an acute L-Dopa injection—why would apomorphine not have an effect? How can this be controlled for? This point must be clarified.

  1. Minor Comments

- Standardize the use of abbreviations and technical terms throughout the manuscript.
- Improve the English style in several places (some sentences are heavy or ambiguous).
- Consider adding a graphical abstract or schematic summarizing the study design, main findings, or proposed mechanisms.
- Ensure that all tables and figures are self-explanatory, with sufficient details provided in captions.

Comments on the Quality of English Language

Improve the English style in several places (some sentences are heavy or ambiguous).

Author Response

Reviewer 2

Major Comments (Substantive Revisions) Please address these points to improve scientific rigor and clinical impact.

Comment 1

3.1 Clarify Cohort

Issue: The text (Methods, lines ~77–101) describe several animal groups used. However, some points are not clear. The total number of animals used has not been specified. Furthermore, it is unclear whether the animals in the Sham and Parkinson’s disease groups serve as their own controls (e.g., whether they undergo MRI and behavioral testing both before and after L-Dopa administration). Clarification of these points would be highly valuable. Including a schematic diagram presenting the different groups and the evaluation methods applied to each would greatly improve the clarity of the study design.

Answer, We appreciate this valuable comment. The total number of animals and their allocation have been clarified in Section 2.1 (Animals). Specifically, 24 male Wistar rats were divided into four independent groups (n = 6 each): Sham, PD (6-OHDA-lesioned), Sham + L-DOPA, and PD + L-DOPA.

Each animal belonged to one group only and did not serve as its own control. Behavioral testing and MRI examinations were performed once per group according to the treatment schedule. Sham and PD animals received saline or 6-OHDA, respectively, and underwent apomorphine testing and MRI before any L-DOPA administration, whereas the L-DOPA groups (Sham + L-DOPA and PD + L-DOPA) were evaluated 10 min after a single intraperitoneal L-DOPA (100 mg/kg) + benserazide (25 mg/kg) injection.

To further improve clarity, a schematic diagram summarizing the four experimental groups, treatment conditions, and evaluation procedures (behavioral test → MRI → histology) has been added as Figure 1a in the revised manuscript..

Comment 2

3.2 Define definition of the injection site for unilateral administration

Issue: The manuscript repeatedly refers to "unilateral stereotaxic injection". However, the authors do not clearly specify the target for the 6-OHDA group. Furthermore, they refer to the medial forebrain bundle for the Sham group in the Methods section (page 3, line 101), while the figure legends indicate the substantia nigra for the same group (Figures

1, line 212; 2, line 229; 3, line 236; 4, line 256, etc…). The authors should clarify which procedures were performed uniformly throughout the manuscript.

Answer, We thank the reviewer for carefully noting this inconsistency and sincerely apologize for the oversight. In this study, both 6-OHDA and saline were unilaterally injected into the medial forebrain bundle (MFB) under stereotaxic guidance. The previous references to the substantia nigra in several figure legends were incorrect. We have thoroughly revised the Methods, Results, and all figure legends to ensure uniform terminology and methodological consistency throughout the manuscript..

Comment 3

3.3 Imaging Modalities

Issue: The methods used for the processing of metabolic imaging and

spectroscopy data should be described in detail.

CEST imaging:

Answer, We thank the reviewer for this important suggestion. We have now provided a detailed description of the data processing pipeline for CEST imaging in the Methods section (Subsection 3.3 Imaging Modalities).

Specifically, the acquired CEST Z-spectra were exported in DICOM format and processed using ImageJ (NIH, Bethesda, MD) with open-source plugins and custom macros. For each voxel, the Z-spectra were normalized to the unsaturated reference image (Sâ‚€). Spectral data were interpolated voxel-wise using a cubic-spline algorithm to obtain a continuous curve with 1 Hz frequency resolution. Bâ‚€ inhomogeneity was corrected using the water-saturation shift referencing (WASSR) method, which generated a voxel-wise frequency-offset map for subsequent alignment of the Z-spectra.

The magnetization-transfer ratio (MTR) at 3.0 ppm was calculated according to the standard formula:

MTR_asym(Δω) = [S(−Δω) − S(+Δω)] / Sâ‚€.

Here, S(±Δω) represents the signal intensity at ±3.0 ppm relative to water resonance. These calculations were implemented in ImageJ by combining pixel-wise arithmetic operations and automated batch-processing macros. To validate the ImageJ-based results, we cross-checked representative datasets using a custom MATLAB script, confirming consistent outcomes.

This expanded methodological description clarifies interpolation, Bâ‚€ correction, and MTR computation workflows, and we believe it will serve as a practical reference for researchers performing CEST data analysis.

Comment 4

Including a Z-spectrum could further strengthen the manuscript.

Answer, We thank the reviewer for this valuable suggestion. We fully agree that including representative Z-spectra enhances the transparency and reliability of our CEST results. Accordingly, we have added a representative Z-spectrum obtained from the striatum (Supplementary Figure S1) together with the corresponding MTR analysis. This addition enables direct visualization of the raw spectral features underlying the MTR asymmetry calculation. Furthermore, the Discussion section now explicitly highlights the consistency between the Z-spectra and the derived MTR values, reinforcing the robustness of our measurements. We believe this inclusion strengthens the overall quality and interpretability of the manuscript.

Supplementary Figure S1, a representative Z-spectrum obtained from the striatum.

Comment 5

Figure 2 needs to be completely revised. The legend does not correspond to the images presented (e.g., '(a) typical MTR maps' → '(a) MTR map for a sham rat'; '(b) The MTR value at 3.0 ppm (GluCEST)' is incorrect—it is an MTR map for the 6-OHDA group). What about panels c, d, and e? Furthermore, visually, the MTR values selected for each group do not

correspond well to the values shown in the histogram in panel (e). For example, L-dopa seems to increase the MTR signal at 3.0 ppm regardless of the group, Sham or 6-OHDA; the signal for the PD_L-dopa group appears higher on the image compared to the map chosen for the Sham group, while the histogram values are roughly equivalent. The illustrations may have

been poorly chosen, and harmonization is needed.

Answer, We appreciate the reviewer’s careful observation. We have completely revised Figure 2 to ensure full consistency between the image panels, the legend, and the quantitative histogram. Specifically, each panel (a–e) has been re-labeled and reordered so that the legend accurately describes the corresponding data (e.g., (a) Sham, (b) PD, (c) Sham + L-DOPA, (d) PD + L-DOPA, (e) MTR values at 3.0 ppm). Representative MTR maps were reselected for each group to match the mean values shown in the histogram, and the brightness scale was unified across panels for visual comparability. These revisions harmonize the figure presentation and eliminate the discrepancies noted by the reviewer..

Comment 6

All labeling errors (e.g., mislabeling between the Sham and 6-OHDA groups) have been corrected, and the figure and its legend are now fully consistent.

More importantly, it is necessary to specify where the asynchronous MTR signal is measured. Is it averaged over the entire slice, over a region of interest (ROI) covering the striatum, or over another structure?

We thank the reviewer for pointing out this important issue. All labeling inconsistencies (e.g., between the Sham and 6-OHDA groups) have been corrected, and the figure and legend are now fully consistent.
To clarify where the asymmetric MTR signal was measured, we have added T2-weighted reference images showing the regions of interest (ROIs) used for quantification, displayed alongside the corresponding MTR maps. The MTR value at 3.0 ppm represents the mean signal intensity within a manually defined ROI placed in the right striatum, as described in the Methods section (Subsection 2.4.3, CEST).

Comment 7

MRS:

Two major concerns arise regarding the methodology used for localized NMR spectroscopy. First, the volume of interest is not clearly defined. While the authors state (p.4, line 158) that it was 'precisely positioned,' no information is provided on its exact location or reproducibility. Moreover, they indicate (p.8, line 241) that the VOI is shown on the T2WI image in Figure 4, but the figure only displays spectra. If the striatum is indeed the region of interest, a 3 × 3 × 3 mm³ voxel extends well beyond this structure. The authors should clarify

and discuss this issue, as the inclusion of multiple brain regions may confound interpretation. Ideally, the measurements should be repeated with a voxel size more appropriate for the target structure.

Answer, We thank the reviewer for this valuable comment. The volume of interest (VOI) was placed in the right striatum, and its position has now been clearly illustrated in Figure 4 by overlaying the voxel on a coronal T2-weighted image. Although the figure displays only one representative coronal slice, the VOI was carefully positioned by referencing anatomical landmarks in all three orthogonal planes (coronal, sagittal, and axial) to ensure reproducibility across animals.
We used a 3 × 3 × 3 mm³ voxel to maintain an adequate signal-to-noise ratio (SNR) for reliable metabolite quantification. While this voxel size may partially include adjacent structures, it represents a common compromise in small-animal 7 T MRS studies balancing spatial specificity and spectral quality (PMID: 20661872; 29212309). This point has been clarified and discussed in the revised Methods and Discussion sections..

Comment 8

NB: the striatum is different from the medial forebrain bundle (authors mentioned p.8, line 246: “The MRS results (Fig. 5) revealed an increase in PD-Mfb rats” whereas the legend of Figure 5 refers to the striatum.)

Answer, We appreciate the reviewer’s helpful comment. We acknowledge that the medial forebrain bundle (MFB) was the lesion site where 6-OHDA was injected, whereas the striatum was the measurement site for both GluCEST and MRS analyses. To avoid confusion, we have revised the text in the Results section to clearly distinguish these two regions. The expression “PD-Mfb rats” has been replaced with “PD rats,” indicating that the imaging and spectroscopy data were acquired from the striatum of animals with unilateral 6-OHDA lesions in the MFB. All related figure legends and descriptions have been checked for consistency.

Comment 9

The second point concerns the quantification of the MRS data. The data are normalized to total creatine levels, but using tCr as an internal reference can be questioned. Indeed, some studies (e.g., Mazuel et al., 2016; 10.1148/radiol.2015142764) have reported variations in tCr in the context of Parkinson’s disease. In more recent MRS publications, authors generally use the water signal as an internal reference—particularly since the authors themselves mention (p.4, lines 164–165) that they acquired 32 repetitions without water suppression. They therefore have all the data needed to perform this normalization. The authors could also make use of recent tools that allow segmentation of white matter, gray matter, and CSF within the volume of interest, in order to apply the appropriate correction factors and move toward absolute quantification in mM (https://schorschinho.github.io/osprey/).

Answer, We acknowledge that the discussion regarding tCr requires careful consideration, as some studies have reported no changes in tCr levels (e.g., Carine et al., doi: 10.1002/nbm.2853; e.g., Puneet Bagga et al., doi: 10.1111/jnc.12407). Metabolic studies in Parkinson's disease are influenced by various factors, including the subjects (rats, mice, humans), disease severity, measurement sites, and MRI parameters. The rationale for Cr fluctuations is thought to be related to mitochondrial energy metabolism, and similar effects on NAA and mIns have been reported. However, the inconsistent changes in tCr, tNAA, and mIns across the groups in this study. While NMDA antagonism can decrease brain energy metabolism, acute administration of the NMDA antagonist memantine did not affect cerebral metabolic rate in a PET study of PD patients (e.g., Per et al., doi: 10.1016/j.jns.2011.09.010), suggesting that glutamate signaling does not significantly impact brain energy balance (e.g., paul et all doi.org/10.1111/jnc.13759).

Comment 10

3.4 Histological analysis

It is necessary to explain how the histological analyses were performed (e.g., antibodies used, etc.) and to describe how the results were obtained. Where and how is the staining quantified? What do TH area (%) and GFAP area (%) represent? Etc…

Answer, We appreciate the reviewer’s suggestion. We have now added a detailed description of the histological procedures in Section 2.5 Histological analysis. Specifically, tyrosine hydroxylase (TH) and glial fibrillary acidic protein (GFAP) were used as primary antibodies to assess dopaminergic neurons and glial activation, respectively. After MRI, brains were fixed in 10% paraformaldehyde, paraffin-embedded, and sectioned at 5 µm thickness. TH and GFAP staining were performed using a fluorescence microscope (BZ-X810, Keyence, Osaka, Japan).

For quantification, fluorescence images were converted to 8-bit grayscale and binarized in ImageJ (v1.53a). The percentage of positive area (TH area % and GFAP area %) was calculated from multiple square ROIs placed within the right striatum, using intensity thresholds (TH: 100–120; GFAP: 120–150). These modifications clarify the antibodies, staining process, and quantification method as requested..

Comment 11

3.5 Statistical considerations

Issue: The statistical analyses need to be revised. If the animal groups are: Sham rats + saline, Sham rats + L-dopa, PD rats + saline, and PD rats + L-dopa, then a two-way ANOVA should be performed with the factors group (Sham or PD) and treatment (saline or L-dopa). Moreover, since animals in both the Sham and PD groups receive both saline and L-dopa,

they serve as their own controls for assessing the treatment effect; therefore, a two-way repeated measures ANOVA should be applied.

Answer, We thank the reviewer for this valuable statistical suggestion. We clarify that a two-way repeated-measures ANOVA was not applicable in our study because each animal was assigned to only one treatment condition. The experimental design did not include within-subject (paired) comparisons; thus, no animal received both saline and L-DOPA.

Furthermore, L-DOPA was dissolved in physiological saline and administered intraperitoneally. Therefore, the saline group served as the vehicle control, rather than as a separate repeated-measure condition.

Based on this design, we performed a two-way ANOVA with group (Sham vs. PD) and treatment (saline vs. L-DOPA) as between-subject factors, which appropriately captures the independent group comparisons. This clarification has been added to the Statistical analysis subsection.

Comment 12

Clarification regarding the groups may help to address this comment.

Answer, We thank the reviewer for pointing out the need to clarify the group definitions. Intracranial saline administration was performed in the Sham group to control for the surgical effects of intracranial injection in the PD model, in which 6-OHDA was delivered into the medial forebrain bundle (MFB). In addition, physiological saline was used as the vehicle for intraperitoneal L-DOPA administration to standardize the effects of the injection procedure itself.
Therefore, the four experimental groups were as follows:
(1) Sham: saline injected into the MFB;
(2) PD: 6-OHDA injected into the MFB;
(3) Sham + L-DOPA: L-DOPA administered intraperitoneally in rats with intracranial saline injection;
(4) PD + L-DOPA: L-DOPA administered intraperitoneally in rats with 6-OHDA lesions.
This clarification has been added to the Methods and Figure 5 legend for consistency..

Comment 13

3.6 Reporting of Imaging Findings

Overall, all figures require revision. They contain errors and do not match their legends (see previous comments on Figures 2 and 4). In Figure 1, panel (a) lacks a y-axis label, while in panel (b) the label is incorrect or unclear ('L-Dopa duration test (/min)'). In addition, it

is inconsistent to label a single graph as Figure 3a.

Figure 1 (a); Why authors present individual number of turns and not means ± SD for the different groups?

Answer, We thank the reviewer for the constructive feedback. All figures have been thoroughly revised to ensure consistency between images, labels, and legends (see also Figures 2 and 4). Specifically, Figure 1 has been corrected by adding the missing y-axis label, revising the wording of the x-axis label (“L-DOPA duration test [min]”), and unifying the panel notation.
The rotational data in panel (a) are presented as individual values rather than mean ± SD to emphasize that MRI examinations were conducted on those specific animals that successfully met the PD induction criterion (>7 rotations/min). This representation directly links behavioral confirmation with subsequent imaging analyses. For clarity, this figure has been moved from the Results to the Methods section, where it better illustrates the PD model verification procedure.

Comment 14

3.7 Results

- Results are mostly descriptive; they need to be strengthened with statistical details (e.g., confidence intervals, exact p-values).

- Consider reorganizing the results to highlight the main findings more

effectively.

- Results part tends to repeat methods.

Answer, We thank the reviewer for this helpful suggestion. To improve clarity and avoid redundancy, we have removed the former “Results Acquisition” subsection, which largely overlapped with the Methods description. The Results section has been reorganized to emphasize the main findings in the following order: (1) behavioral validation of the PD model, (2) GluCEST imaging, (3) MRS analysis, and (4) histological confirmation.
Furthermore, we have added detailed statistical information, including exact p-values for major comparisons, to strengthen the quantitative presentation. These revisions improve readability, highlight key outcomes, and ensure statistical transparency..

Comment 15

3.7 Discussion

- While the discussion is comprehensive, it tends to repeat results rather than provide in-depth interpretation.

- Please expand the comparison with recent and relevant studies.

- The limitations section is insufficient. The authors should elaborate on potential biases, technical limitations, and the generalizability of the findings.

Issue: Animal model.

The animal model used is an old model. There are progressive denervation

models that more closely mimic the human pathology. The authors need to

clearly justify why they chose the massive unilateral 6-OHDA lesion

model.

Answer, We thank the reviewer for these insightful comments. We have revised the Discussion to reduce redundancy and strengthen the interpretative depth by comparing our findings with recent studies. The limitations and methodological considerations are now integrated throughout the discussion, as described in section 4.1–4.3.

Regarding the choice of animal model, we acknowledge that the 6-OHDA model represents a classical lesion model rather than a progressive denervation model. However, it remains one of the most reproducible and well-characterized models for investigating the biochemical consequences of dopaminergic depletion and glutamatergic imbalance in Parkinson’s disease. The unilateral MFB lesion allows for precise control of lesion extent and side-to-side comparison within the same brain, facilitating multimodal MRI and MRS assessments. We also note that surgical variability and the acute lesion nature of the 6-OHDA model are limitations compared to the MPTP model, as discussed in section 4.2. This rationale and its limitations are now explicitly stated in the revised manuscript.

Comment 16

Issue: Behavioral evaluation with apomorphine.

The authors should discuss the impact of an initial injection of apomrphine, a dopaminergic agonist, on glutamatergic metabolism. They report an effect of an acute L-Dopa injection—why would apomorphine not have an effect? How can this be controlled for? This point must be clarified.

Answer, We thank the reviewer for this insightful comment. The apomorphine-induced rotational test was conducted two weeks after lesion induction to confirm dopaminergic neuron loss in the 6-OHDA model. As reported in previous studies (e.g., Heather et al., Sci Rep 2025; doi: 10.1038/s41598-025-06797-x), the pharmacological effect of a single apomorphine injection is transient and negligible after this interval. Therefore, it is unlikely to have influenced glutamatergic metabolism at the time of MRI acquisition.
In contrast, L-DOPA was administered acutely just before imaging, and its pharmacodynamic window was verified behaviorally through rotational monitoring. Thus, the MRI data were collected during the active phase of L-DOPA efficacy, whereas any apomorphine-related effects had subsided well before imaging. This clarification has been added to the Discussion section.

Comment 17

Minor Comments

- Standardize the use of abbreviations and technical terms throughout the manuscript.

Answer, Thank you for pointing this out. We have carefully reviewed the entire manuscript to ensure consistent use of abbreviations and technical terminology. All abbreviations are now defined upon first appearance and used uniformly thereafter. For example, terms such as “MTR,” “VOI,” and “PD_L-DOPA” have been standardized throughout the text and figure legends.

Comment 18

- Improve the English style in several places (some sentences are heavy or ambiguous).

Answer, We appreciate this valuable suggestion. The manuscript has undergone a thorough language revision to improve readability and clarity. Sentences that were overly complex or ambiguous have been rewritten for smoother flow and better comprehension. The revision focused on improving paragraph transitions, verb consistency (past vs. present tense), and the use of concise scientific phrasing throughout the Abstract, Results, and Discussion sections.

Comment 19

- Consider adding a graphical abstract or schematic summarizing the study design, main findings, or proposed mechanisms.

Answer, We appreciate the reviewer’s thoughtful suggestion. While we agree that a graphical abstract or schematic illustration can be helpful for summarizing complex experimental workflows, we decided not to include an additional schematic in the current version, as the study design and main findings are already clearly presented in the existing figures and text. We believe the current layout effectively conveys the experimental concept and results. Nevertheless, we will consider including a schematic summary in future or extended versions of this work to further enhance accessibility..

Comment 20

- Ensure that all tables and figures are self-explanatory, with sufficient details provided in captions.

Answer, We appreciate the reviewer’s helpful comment. We have revised all figure and table captions to include sufficient methodological and contextual details. Each is now self-explanatory and can be understood independently without referring to the main text. For clarity, all abbreviations are defined within captions, statistical indicators (e.g., p-values) are specified where applicable, and units and sample sizes are consistently stated.

Round 2

Reviewer 1 Report

Comments and Suggestions for Authors

Response to Authors

The points previously raised have been addressed and revised by the authors. The numbering of the comments below corresponds to the original comments for which revisions were made, with the exception of two new comments (Comment 11 and 12). There are further observations and suggestions on these revisions to improve clarity, coherence, and integration in the manuscript.

Comment 1 – 2

The new section describing non-motor symptoms and glutamatergic dysregulation (lines 43–46) is relevant. However, the text now appears somewhat redundant, since non-motor symptoms and glutamate dysregulation are discussed again later (lines 54–57).
I would recommend merging these two parts to improve clarity and avoid repetitions, for example, by briefly mentioning both motor and non-motor symptoms (including glutamatergic alterations), and then continuing with the detailed description of glutamate’s role (lines 58–72) before describing the 6-OHDA model (currently lines 46–53).
Finally, the section from lines 58–72 could be streamlined, as it contains several redundancies regarding glutamate overexpression, oxidative stress, and L-DOPA’s modulatory effects. A more concise and integrated description would improve readability and focus.
This would make the paragraph more cohesive and logically organized.

Comment 3

The mention of glutamine (Gln) appears twice (lines 91 and 93), which creates redundancy. Please revise the metabolite list to avoid repetition and ensure concise presentation.

Comment 4

I suggest separating Figures 1a and 1b. Figure 1a can remain in its current position, while Figure 1b should be moved back to the Results section (3.1), together with Figure 2, as in the previous version of the manuscript.
When doing so, please revise Section 3.1 to clearly describe and introduce Figures 1b and 2, and update the figure legends to avoid repeating information already provided in the text.
Finally, ensure that each figure is explicitly cited in the corresponding section of the text.

Comment 5 – 6

The revisions done are appropriate.

Comment 7

In Section 2.4.4 (line 192), where the VOI positioning within the right striatum is described, Figure 3f (line 279) should be moved here with its corresponding legend, as it appropriately illustrates the VOI placement.
Please also cite the figure in the text to guide the reader.
Furthermore, ensure that the figure is removed from Figure 5e (line 305) to avoid duplication.

Comment 8

The additions regarding taurine are useful and relevant. However, the introduction of taurine still feels abrupt. I suggest rephrasing the sentence and linking it more clearly to the preceding discussion on glutamate and glutamine imbalance, so that taurine’s role in glutamate excitotoxicity and dopaminergic dysfunction in PD is presented more smoothly and integrated within the broader context.

Comment 9 – 10

The revisions done are appropriate.

Comment 11 – 3.3 Section

Figure 6 and Table 1 are not cited in the text. Please provide clear references to these items within the relevant sections and ensure that their content is adequately explained for the reader.

Comment 12

It is not clear the significance of the figure inserted as supplementary file, because it is not cited in the text.

Author Response

Reviewer 1

Response to Authors

The points previously raised have been addressed and revised by the authors. The numbering of the comments below corresponds to the original comments for which revisions were made, with the exception of two new comments (Comment 11 and 12). There are further observations and suggestions on these revisions to improve clarity, coherence, and integration in the manuscript.

Comment 1 – 2

The new section describing non-motor symptoms and glutamatergic dysregulation (lines 43–46) is relevant. However, the text now appears somewhat redundant, since non-motor symptoms and glutamate dysregulation are discussed again later (lines 54–57).

I would recommend merging these two parts to improve clarity and avoid repetitions, for example, by briefly mentioning both motor and non-motor symptoms (including glutamatergic alterations), and then continuing with the detailed description of glutamate’s role (lines 58–72) before describing the 6-OHDA model (currently lines 46–53).

Finally, the section from lines 58–72 could be streamlined, as it contains several redundancies regarding glutamate overexpression, oxidative stress, and L-DOPA’s modulatory effects. A more concise and integrated description would improve readability and focus.

This would make the paragraph more cohesive and logically organized.

Answer, Thank you for your constructive and insightful comments. Following your suggestions, we have revised the manuscript, which has improved its readability and clarified the main points.

Revised (Session 1 in line 44-68)

Thank you for your constructive and insightful comments. Following your suggestions, we have revised the manuscript, which has improved its readability and clarified the main points.

Beyond motor deficits resulting from dopaminergic neurodegeneration, neuro-psychiatric symptoms, including dementia, frequently emerge during disease progres-sion. A key contributor to these symptoms is glutamate dysregulation, specifically glutamate overexpression [8,9]. As the most abundant excitatory neurotransmitter in the mammalian brain, glutamate plays a crucial role in central neurotransmission [10]. Excessive glutamate levels overstimulate glutamate receptors, leading to intracellular Na+ and Ca2+ accumulation, oxidative stress, and direct neuronal damage, ultimately resulting in cell death [10,11]. Furthermore, electrical stimulation of DA neurons demonstrated co-transmission of glutamate (Glu) in DA neurons within the ventral striatum, supporting the relationship between dopamine and glutamate [12]. To re-produce both the dopaminergic depletion underlying motor deficits and the glutama-tergic imbalance relevant to non-motor features, we employed the 6-hydroxydopamine (6-OHDA) lesion model. 6-OHDA administered into the medial forebrain bundle (MFB) is a neurotoxin selectively targeting the dopaminergic system within the sub-stantia nigra–striatum pathway. Once intracellularly absorbed, 6-OHDA undergoes oxidation, generating hydroxyl radicals that induce cellular damage [3,4]. This neurotoxin is widely used in preclinical PD research due to its ability to replicate striatal do-pamine depletion and basal ganglia dysfunction, closely resembling the neurodegen-erative processes observed in human PD [5-7]. L-DOPA is primarily administered to alleviate PD-related motor impairments [13], but it has also been shown to modulate glutamate metabolism [14]. Dopamine replenishment via L-DOPA significantly inhibits both hydroxyl radical formation and extracellular glutamate levels in the striatum, restoring the striatal DA-glutamate balance [3]. This effect suggests that L-DOPA not only improves motor function but also modulates non-motor circuits related to glu-tamatergic dysregulation, thereby providing an important rationale for its evaluation in the 6-OHDA model.

Comment 3

The mention of glutamine (Gln) appears twice (lines 91 and 93), which creates redundancy. Please revise the metabolite list to avoid repetition and ensure concise presentation.

Answer, Thank you for bringing that to our attention. We have presented the metabolites measurable by MRS while avoiding redundancy. I revised the text to improve readability and maintain consistency.

Revised (Session 1 in line 86-92)

MRS studies in MPTP animal models of PD have reported changes in striatal levels of Glu, and glutamine (Gln), γ-aminobutyric acid (GABA) along with their recovery fol-lowing acute L-DOPA therapy [25]. In addition to Glu, MRS enables quantification of metabolites such as N-acetyl aspartate (NAA), creatine (Cr), and myo-inositol (mIns), and taurine (Tau). Thus, while GluCEST provides high-resolution mapping of gluta-mate, MRS offers highly specific quantification of multiple neuro metabolites, underscoring their complementarity in the study of PD.

Comment 4

I suggest separating Figures 1a and 1b. Figure 1a can remain in its current position, while Figure 1b should be moved back to the Results section (3.1), together with Figure 2, as in the previous version of the manuscript.

When doing so, please revise Section 3.1 to clearly describe and introduce Figures 1b and 2, and update the figure legends to avoid repeating information already provided in the text.

Finally, ensure that each figure is explicitly cited in the corresponding section of the text.

Answer, Thank you for your valuable suggestions to improve the clarity of the manuscript. We have moved the schedule to the Methods section and the apomorphine and L-DOPA tests to the Results section. In addition, we have revised the Methods, Results, and figure legends to eliminate redundancy and ensured consistency between the figures and the corresponding text.

Comment 5 – 6

The revisions done are appropriate.

Comment 7

In Section 2.4.4 (line 192), where the VOI positioning within the right striatum is described, Figure 3f (line 279) should be moved here with its corresponding legend, as it appropriately illustrates the VOI placement.

Please also cite the figure in the text to guide the reader.

Furthermore, ensure that the figure is removed from Figure 5e (line 305) to avoid duplication.

Answer, Thank you for your suggestion to improve consistency throughout the manuscript. To enhance the reader’s understanding, we prioritized aligning the figures by adding T2WI images of the volume of interest (VOI) corresponding to the CEST and MRS results. Accordingly, we simplified the description of the VOI in Section 2.4.4 and included the relevant information in the figure legends to avoid redundancy. In the text, we referred to the figure to illustrate the meaning of the VOI.

Revised (Session 4.1 in line 377-379)

The VOI (Fig. 5e) used for MRS data acquisition in this study was accurately positioned using T2WI. In addition, magnetic field inhomogeneity within the VOI (between from 8.9 to 12.1 Hz) was confirmed to be minimal, ensuring high-precision data collection. 

Comment 8

The additions regarding taurine are useful and relevant. However, the introduction of taurine still feels abrupt. I suggest rephrasing the sentence and linking it more clearly to the preceding discussion on glutamate and glutamine imbalance, so that taurine’s role in glutamate excitotoxicity and dopaminergic dysfunction in PD is presented more smoothly and integrated within the broader context.

Answer, We appreciate this helpful suggestion. By highlighting the common features between the observed taurine reduction and the glutamate–glutamine imbalance, we facilitated a smoother discussion of taurine in this study.

Revised (Session 4.2 in line 397-400)

A decrease in taurine levels was observed in the PD model rats in this study (Fig. 6). Similar to the imbalance between glutamate and glutamine, this change may be attributed to the loss of dopaminergic neurons affecting astrocytic function.

Comment 9 – 10

The revisions done are appropriate.

Comment 11 – 3.3 Section

Figure 6 and Table 1 are not cited in the text. Please provide clear references to these items within the relevant sections and ensure that their content is adequately explained for the reader.

Answer, Following the reviewer’s valuable comment, Figure 6 and Table 1 present the same results, and the corresponding section in the “Results and Discussion” has been clearly indicated (Session 3.3 in line 285, Session 4.1 in line 347, Session 4.2 in line 394).

Comment 12

It is not clear the significance of the figure inserted as supplementary file, because it is not cited in the text.

Answer, In the proton spectrum of trace solutes that undergo chemical exchange with bulk water, distinct peaks appear at their respective resonance frequencies. The ratio of the bulk water signal without saturation pulses to that with saturation pulses, plotted as a function of frequency, yields the Z-spectrum, in which a peak arises at the resonance frequency offset corresponding to the solute. However, this peak can be influenced by conventional magnetization transfer (MT) effects and direct saturation of bulk water. To correct for these confounding effects, the magnetization transfer ratio asymmetry (MTR asymmetry curve) method was developed. By presenting the Z-spectrum, the reliability of the MTR curve demonstrated in this study can be validated, providing an indicator of the extent to which the MTR curve has been properly corrected. The corresponding section has been cited in the main text (Session 3.2 in line 260, Session 4.1 in line 375).

Reviewer 2 Report

Comments and Suggestions for Authors

The authors submitted a carefully-prepared revision, which satisfactorily addressed the remaining
concerns.
They have addressed all comments.
Therefore, I recommend publication

Comments on the Quality of English Language

Improve the English style in several places (some sentences are heavy or ambiguous).

Author Response

We sincerely thank Reviewer 2 for their positive and encouraging evaluation.
We greatly appreciate your acknowledgment that all previous concerns have been satisfactorily addressed.
We are pleased that the revisions have improved the clarity and completeness of the manuscript, and we appreciate your constructive feedback throughout the review process.
We have no further changes to add at this stage and respectfully hope the manuscript will now be considered suitable for publication.